# Environmental Changes Recorded in Tufa from the Korana River, Croatia: Geochemical and Isotopic Approach

Andreja Sironić [1,*,†], Mavro Lučić [1,†], Igor Felja [2] and Darko Tibljaš [2]

1 Ruđer Bošković Institute, 10000 Zagreb, Croatia
2 Department of Geology, Faculty of Science, University of Zagreb, 10000 Zagreb, Croatia
* Correspondence: andreja.sironic@irb.hr
† These authors contributed equally to this work.

**Abstract:** Between 1979 and 2003, a 35 cm thick layer of laminated tufa formed around a wooden pillar located at the beginning of the Korana River, which emerges from the Plitvice Lakes, Croatia. The laminated tufa structure allowed for the stratigraphic analysis of mineral, elemental and $\delta^{18}O$ and $\delta^{13}C$ isotope composition. Results are compared to other carbonate archives (lake sediments) from the Plitvice Lakes and to measured water physical–chemical data collected in the same time. An increase in water temperature and decrease in Mg/Ca ratio was observed during that period. Results showed that the tufa Mg/Ca ratio and $\delta^{18}O$ could not be used as a temperature proxy, while an observed $\delta^{18}O$ increase after 1990 was attributed to increase in evaporation rate. Furthermore, the constant increase in redox-sensitive elements (Mn) towards the external part of tufa may indicate periodic events of anoxia at the sediment/water interface. Local enrichment factors (LEFs) for potentially toxic elements were calculated using the background function and Al as the normalizing element. The LEFs showed an enrichment pattern of Cu and Pb in 1983 and 2001, which could indicate their anthropogenic origin and the sensitivity of the tufa to environmental changes.

**Keywords:** tufa carbonate; $\delta^{18}O$; $\delta^{13}C$; elemental composition; physical proxies; anthropogenic influence; Plitvice Lakes





## 1. Introduction

It is believed that the world's karst accounts for about 21% of unknown continental carbon sinks [1], so understanding processes in the karst is important. Karst is developed from carbonates (limestone and dolomite), interspersed with cracks, holes, channels and caves, sometimes filled with water. Karst water is typically oversaturated with calcium carbonate, and in places with strong water flow, increased bioactivity and warm conditions, $CaCO_3$ precipitates in the form of tufa or lake sediment [2–5]. In the Dinaric karst area of Croatia, the Plitvice Lakes are formed from a system of barrages made of bioinduced tufa. Due to tufa and the karst lakes sediment sensitivity to environmental changes, the Plitvice Lakes system has been a subject of studies on numerous occasions and is regarded as nature's own laboratory, e.g., [2,6–9].

Tufa is a porous, often soft, variety of limestone formed by the physical and bioinduced precipitation of calcium carbonate from hard water [6]. It is usually formed on the waterfalls of rivers and lakes. Tufa calcite is precipitated in association with a biofilm, a by-product of the microbial metabolic activity of diatoms, bacteria and/or cyanobacteria and also contains remains of micro- and macrophytes and bacteria [7,10–12]. Due to its biogenic characteristics, the process of tufa precipitation is very sensitive to changes in water physical, chemical or biological properties and to seasonal fluctuations, e.g., temperature change [2,13]. Tufa deposited in layers can be used as an archive of fast changes in, for example, chemical, hydrological or biological conditions (e.g., [14]), giving higher precision and accuracy compared to, e.g., sediments that require radiometric dating.

In the period from 1981 to 2014, an increase in temperature was observed in most of the waters of the Plitvice Lakes [8]. Higher temperatures may increase the decomposition of organic matter (OM), resulting in higher $CO_2$ concentration ($pCO_2$) and lower pH values, possibly reversing the process of calcification in the lakes. Although increased $pCO_2$ does not reduce the potential for photosynthetically induced calcification in microbial films, high $pCO_2$ can lower pH and reduce the potential for the physicochemical precipitation of calcium carbonates [10]. With greater OM degradation, the concentration of dissolved organic carbon (DOC) also increases and acts as a tufa growth inhibitor, since DOC molecules form chelate complexes around $Ca^{2+}$ ions [2,15,16].

Another consequence of temperature increase is of a physical nature and is expressed as a decrease in oxygen concentration. Oxygenation, i.e., redox conditions, can be studied from Mn/Ca, or Co×Mn values [17]. Concentrations of uranium, vanadium, molybdenum and to a lesser extent certain other trace metals such as chromium and cobalt can be used as paleoredox proxies [18].

In carbonates from marine sediments and shells, as well as from laminated tufa and speleothems, temperature changes can be detected using proxies such as Mg/Ca [19–23] or $\delta^{18}O$ [24–26]. Sr/Ca is also often used as a proxy for temperature in corals and shells [27–31]. $\delta^{18}O$ can be used as a temperature proxy in equilibrium conditions of carbonate precipitation. However, it can also often be linked to high evaporation rates, so it is important to determine whether carbonate precipitation occurs under equilibrium conditions when $\delta^{18}O$ is used as a temperature proxy [14,25]. The temperature effect on $\delta^{18}O$ is often overridden by evaporation, and Mg/Ca as a temperature indicator in tufa is only relevant in low ionic strength solutions with low Mg/Ca ratios (e.g., [19]) and is sometimes unreliable [26,32,33] because it can be hindered by water chemistry and detrital material such as dolomite that may be found in tufa. A further problem with temperature proxies such as $\delta^{18}O$ and Mg/Ca is that they are connected only with carbonate precipitation, and since the tufa does not grow during winter seasons, they represent only warmer periods. At the springs, lakes and river of the Plitvice Lakes, a decrease in Mg/Ca was observed from periods 1980–1984 to 2010–2014, while Mg/Ca had strong correlation with temperature [8].

The composition of $^{13}C$ is a useful tool for studying carbonate sediments such as tufa. Secondary carbonates contain carbon of both geogenic (from limestone carbonate) and biogenic origin. The isotope composition ($\delta^{13}C$) of secondary carbonates points to the origin of carbon, to processes involved during the material formation, and can give insight into changes in the environment (e.g., [25,34–40]). Tufa and lake sediment do not precipitate in equilibrium conditions, as is the case in speleothem, so physical–chemical conditions, such as temperature and evaporation rate, can also play a role in $\delta^{13}C$ values [36].

There are numerous studies on the elemental composition of the surrounding soils [41], sediments [42] and water samples of the Plitvice Lakes [43,44]. These studies have shown that the variability in elemental composition is mainly caused by natural processes, with lithology playing the most important role, regardless of the environmental compartment. However, in some studies focused on lake sediments, anthropogenic activity was found in lakes Prošće, Kozjak and Kaluđerovac [45,46]. Similar pollution trends for Pb and Sb were interpreted as a result of alkyl-Pb compounds used as anti-knock additives for gasoline in cars from the 1950s to the 2000s. For the other metals, the variations were in the range of the background values, indicating an environment with minimal anthropogenic pressure [45,46].

This study investigates the elemental composition and composition of oxygen and carbon isotopes in 24-year-old laminated tufa that had grown around a wooden pillar at the outlet of Plitvice Lakes from 1979 to 2003 [47,48]. The main issue that initiated this research was the contradiction between the observed temperature increase and a three-decade Mg/Ca decrease in waters of the Plitvice Lakes, where Mg/Ca is considered a temperature proxy. Elemental analysis of the tufa together with another temperature proxy, $\delta^{18}O$, have the potential to explain this contradiction. The approach of this research was (i) to observe the physical and chemical changes during the tufa growth and (ii) to link

these changes with known hydrological and metrological observations that define the changes in the environmental system of the Lakes, while also using Mn/Ca as oxygen concentration proxy and $\delta^{13}C$ and C/N as bioproductivity proxies. While defining the elemental characteristics of the tufa, the research also discusses anthropogenic influences on the variation of potentially toxic elements.

## 2. Materials and Methods

### 2.1. Site Description and Geological Setting

Plitvice Lakes National Park is located in the Dinaric karst in the Central Croatia, between mountains Kapela and Plješivica (44°53′ N, 15°37′ E) (Figure 1). The region has a typical continental climate with a mean annual temperature of 9.2 ± 0.5 °C and precipitation between 1148 and 2113 mm (from 1986 to 2019 [49]). The heart of the national park is the Plitvice Lakes system. It consists of 16 cascade lakes separated by waterfalls and tufa barrages at altitudes between 700 m and 500 m a.s.l. (Figure 1). The lakes receive water from two main streams (the Bijela Rijeka and the Crna Rijeka Streams) and two tributaries (the Rječica and the Plitvica Stream). The Plitvice Lakes feed the Korana River (Figure 1) downstream.

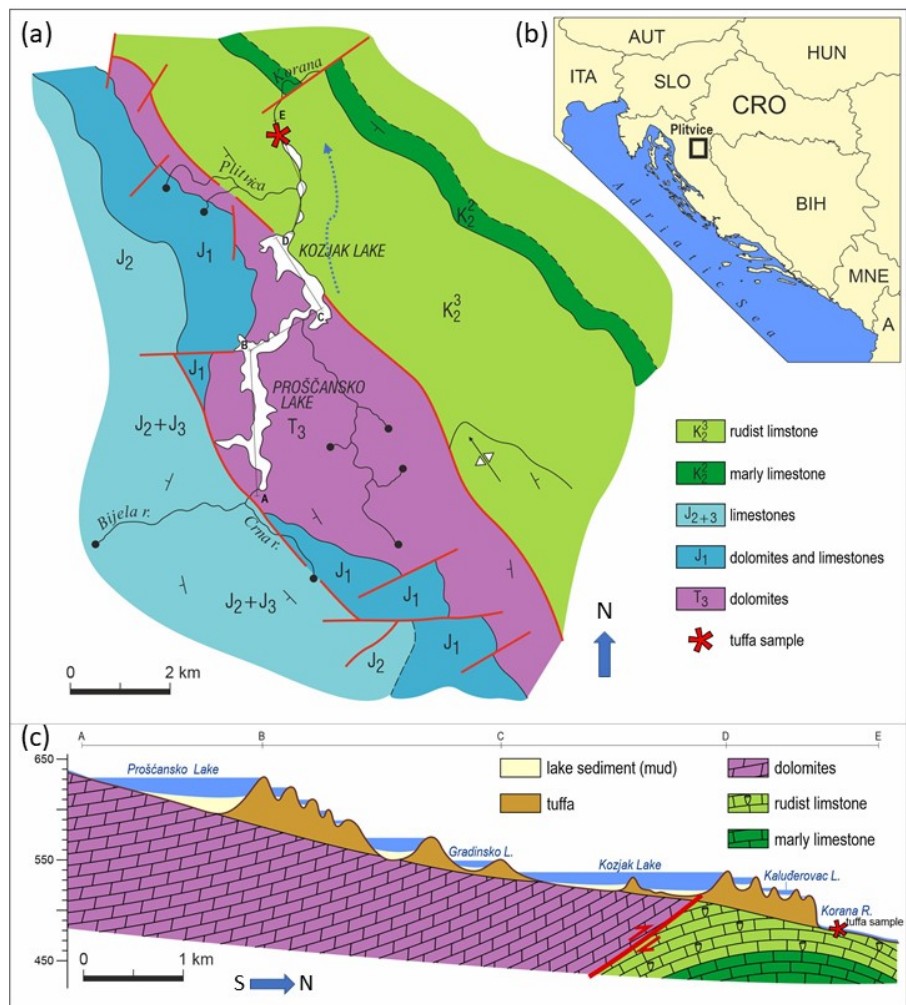

**Figure 1.** (**a**) Geological map of the Plitvice Lakes area (modified after [50]) with the location of the analyzed tufa (marked with red star) and dashed arrow indicating the flow direction, (**b**) setting of the Plitvice Lakes in Croatia. Croatia is bordered by Italy (ITA), Slovenia (SLO), Hungary (HUN) and Bosnia and Herzegovina (BiH), (**c**) geological profile through Upper and Lower Lakes, with clear distinctions between lithology of the basement rocks (modified after [51], image created by Robert Koščal).

In the area of Plitvice Lakes during the Mesozoic (Late Triassic to Late Cretaceous), a 3500–4000 m thick continuous series of carbonates were deposited [50]. The lithological characteristics and tectonic fabric of the terrain influenced the formation of surface and underground flows. Lakes were divided into the Upper and the Lower Lakes by a large fault (Figure 1) [51], which strikes northwest–southeast along the northeast edge of the Lake Kozjak. The lakes developed in the karstic river valley that had been dammed by tufa barriers. The Upper Lakes, to which the two largest lakes, Lake Prošće and Lake Kozjak (Figure 1) belong, were developed in the open part of the river valley composed of dolomite of the Late Triassic age, which has a low permeability and low primary porosity [50]. However, the Lower Lakes (which Lake Kaluđerovac belongs to, shown in Figure 1) were formed in the Upper Cretaceous from well-karstified limestone with a number of caves in the canyon flanks and thus the existence of these lakes is not completely explained. A plausible explanation is that a series of impermeable marly (clayey) limestone in the base of the Upper Cretaceous limestone represents a barrier for underground flow, so the rocks above are completely saturated with water. Therefore, the lithology could explain the existence of the surface water flow. The hydrogeological and lithological characteristics of the basement rocks, water that shaped the relief, tectonics and formation of the tufa barriers played the key role in the formation of the lakes [51].

The Plitvice Lakes are hard-water lakes with conditions favorable for calcite precipitation (range of saturation index, ISAT, $CaCO_3$ 3–10 and pH 8.0–8.7) [2,52]. Calcite precipitation rate is the greatest between spring and autumn, with lower precipitation rates in wintertime [2]. Precipitation of carbonates does not occur at springs and some tributaries.

Tufa, analyzed in this paper, was located in the area built of limestone of the Late Cretaceous age exactly at the beginning of the Korana River. This limestone is very porous and permeable rudist limestone, characterized by many caves, pits, cracks and other karstic forms.

### 2.2. Sampling and Mineralogical, Isotope and Chemical Analyses

The analyzed tufa grew around a wooden pillar (Figure 2), a load-bearing column of a bridge which was placed in water stream of the Korana River in 1979 [47]. When the pillar was retrieved from the river stream in 2003 it was surrounded by a 35 cm thick layer of tufa. The layer was subsampled in 2008 (Figure 2) for mineral, trace element and $\delta^{13}C$ and $\delta^{18}O$ isotope analyses. Ten layers were subsampled following the tufa growth line, using scalpels and spatulas in an amount of 65–75 g of sample per layer. The samples were taken precisely with a stainless steel scalpel and spatula in a sterile laboratory environment. Tools without copper, antimony or lead were used in sampling to prevent sample contamination. Each layer was homogenized and assigned to a date by approximating the linear tufa growth from 1979 to 2003 (Table 1) [48].

The tufa was friable, powder-like and highly porose, with uneven growth around the wooden pillar in the middle and worn down on its edges. The layers were well defined up to 6 cm from the pillar, with each layer having a width of 2–3 mm. Samples labelled 6–10 are at positions where the tufa was more worn down and layers were less defined.

Mineral composition of the tufa was determined on 10 previously mentioned samples (Table 1) using the X-ray powder diffraction method (XRPD) and a Philips instrument, X-Pert type, equipped with vertical goniometer, Cu-tube (40 kV, 40 mA, Ni filter), and a PIXcel[1D] detector, before measurement samples were additionally powdered in an agate mortar. Phase analysis and unit cell dimensions of the calcite were determined using Profex software [53]. The equations of Mackenzie et al. [54], which describe a relationship between the dimensions of the unit cells and the magnesium content of Mg calcites, were used to determine the Mg content of the calcites analyzed.

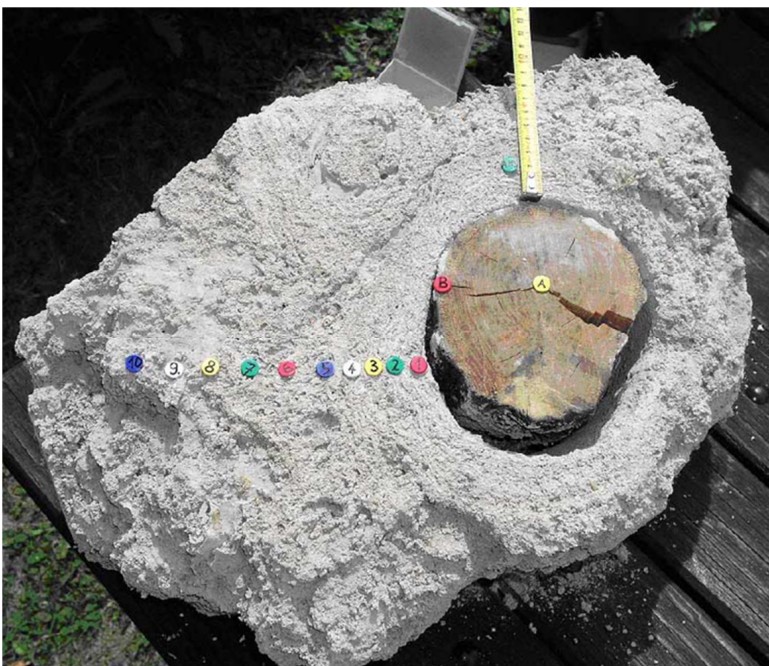

**Figure 2.** Tufa layer around the wooden pillar taken from the Korana River. Labels from 1 to 10 mark places of tufa sub-sampling. (Letters A and B refer to inner and outer wood layers).

**Table 1.** Labels of sampled layers, distance from the beginning of the tufa growth and year assigned to each layer, assuming linear tufa growth.

| Sample Label | Distance from the Beginning of the Growth (cm) | Assigned Year |
|:---:|:---:|:---:|
| 1 | 0–1.5 | 1980 ± 1 |
| 2 | 1.5–2.8 | 1982 ± 1 |
| 3 | 2.8–4.0 | 1983 ± 1 |
| 4 | 4.0–5.3 | 1985 ± 1 |
| 5 | 5.3–6.8 | 1987 ± 1 |
| 6 | 6.8–9.3 | 1989 ± 1.5 |
| 7 | 9.3–11.5 | 1992 ± 1.5 |
| 8 | 11.5–13.7 | 1995 ± 1.5 |
| 9 | 13.7–16.0 | 1998 ± 1.5 |
| 10 | 16.0–19.0 | 2001 ± 2 |

The element analysis was performed via high resolution inductively coupled plasma mass spectrometry (HR ICP-MS; Element 2, Thermo, Karlsruhe, Germany). Before analysis, the samples were digested in a microwave oven (Multiwave ECO, Anton Paar, Graz, Austria) by means of a mixture of acids: 4 mL $HNO_3$ (65%, pro analysis, Kemika) and 2 mL HCl (37%, VLSI Grade, Rotipuran). Since tufa consists mainly of carbonate, the results of this type of digestion do not differ from the total concentrations obtained when HF is used [45]. After digestion, the solution was transferred to a volumetric flask and appropriately diluted with Milli-Q water to optimal concentration levels for ICP-MS measurements. Quality control was provided via simultaneous measurements of the blank and certified reference material. The accuracy of the results was 5–10%, depending on the element measured. Details of the analytical method were described in Fiket et al. [55].

The stable isotope composition, $\delta^{13}C$ and $\delta^{18}O$, were measured at the Stable Isotope Laboratory, Physics Department, School of Medicine, University of Rijeka, Rijeka, Croatia, using a modified method of McCrea [24] on a Delta XP mass spectrometer, with a periphery device Gas Bench [56]. In each run, three international standards (NBS-19, IAEA-CO-1 and IAEA-CO-9) were included for normalization purposes. The resulting carbon isotope ratios

were reported in the delta ($\delta$) notation as per mil (‰) deviation relative to the Vienna Pee Dee Belemnite Standard (V-PDB):

$$\delta^n X = \frac{R_{sample} - R_{standard}}{R_{standard}} \tag{1}$$

where R denotes the ratio of the heavy isotope to the light isotope. The analytical precision was better than $\pm 0.1$‰.

For CHN analyses in organic matter, samples were treated with $1 \, mol \, l^{-1}$ HCl. Dried carbonate-free residues were combusted in a PerkinElmer Series II CHNS/O Analyzer. Atomic C/N ratios of the residue material were calculated from the measured carbon and nitrogen concentrations.

### 2.3. Statistical Analysis

Besides absolute element concentrations, geochemical data should also be considered in terms of their composite nature. This means that each variable is part of a whole and carries only relative information [57]. To follow this definition, before calculating the Pearson product-moment correlation coefficients, we transformed the geochemical data with a special type of log-ratio transformation introduced by Kynčlova et al. [58], called symmetric coordinates (balances). For a better visualization of data structure and processes in tufa, the correlation matrix of symmetric balances as a heatmap is presented. Additionally, groups of elements were clustered using hierarchical clustering based on Ward's method. The analysis was performed with the "gplots" and "robcompositions" packages in statistical programming language R [59].

### 2.4. Local Enrichment Factor Calculation

To determine the potential anthropogenic impact on the elemental composition of tufa samples, the local enrichment factor (LEF) was calculated using the following formulae:

$$LEF = E/E_{BN} \tag{2}$$

$$E_{BN} = f(E_{REF}) \tag{3}$$

The LEF can be obtained through the empirical background function $f(E_{REF})$, where the concentration of the element of interest (E, possibly an anthropogenic element) is plotted against a concentration of a "conservative" lithogenic element ($E_{BN}$) using the least trimmed squares (LTS) regression. LTS is a robust statistical approach where the outliers do not necessarily affect the estimates of the model parameters, allowing for a more accurate detection of the geochemical background with statistical rigor. For more details, see [60].

## 3. Results

### 3.1. Mineralogical Analyses

The XRPD patterns of all analyzed tufa samples are almost identical. In all the samples, calcite is the dominant phase (98–99 wt.%) while quartz (0.5–1.6 wt.%) and dolomite (0.4–1.0 wt.%) are present in trace amounts. A representative powder diffraction pattern, characterized by sharp calcite diffraction peaks, is shown in Appendix A, Figure A1.

According to the calculated calcite unit cell dimensions and the equations of Mackenzie et al. [54], calcite has a stable and very low magnesium content (up to 1.5 mol%) in all analyzed samples.

### 3.2. Multi-Elemental, Isotope and C/N Analyses

The metals for the elemental analyses of tufa are selected to represent different possible inputs, terrestrial or authigenic and anthropogenic. The mass concentrations of metals (Li, Rb, Cs, Mg, Ca, Sr, Ba, U, V, C, Mn, Fe, Co, Zn, Cd, Al, Pb) along with $\delta^{13}$C, $\delta^{18}$O and C/N ratio were measured in each tufa layer which can be assigned to the year of formation from Table 1 (Appendix A, Tables A1 and A2, Figure 3).

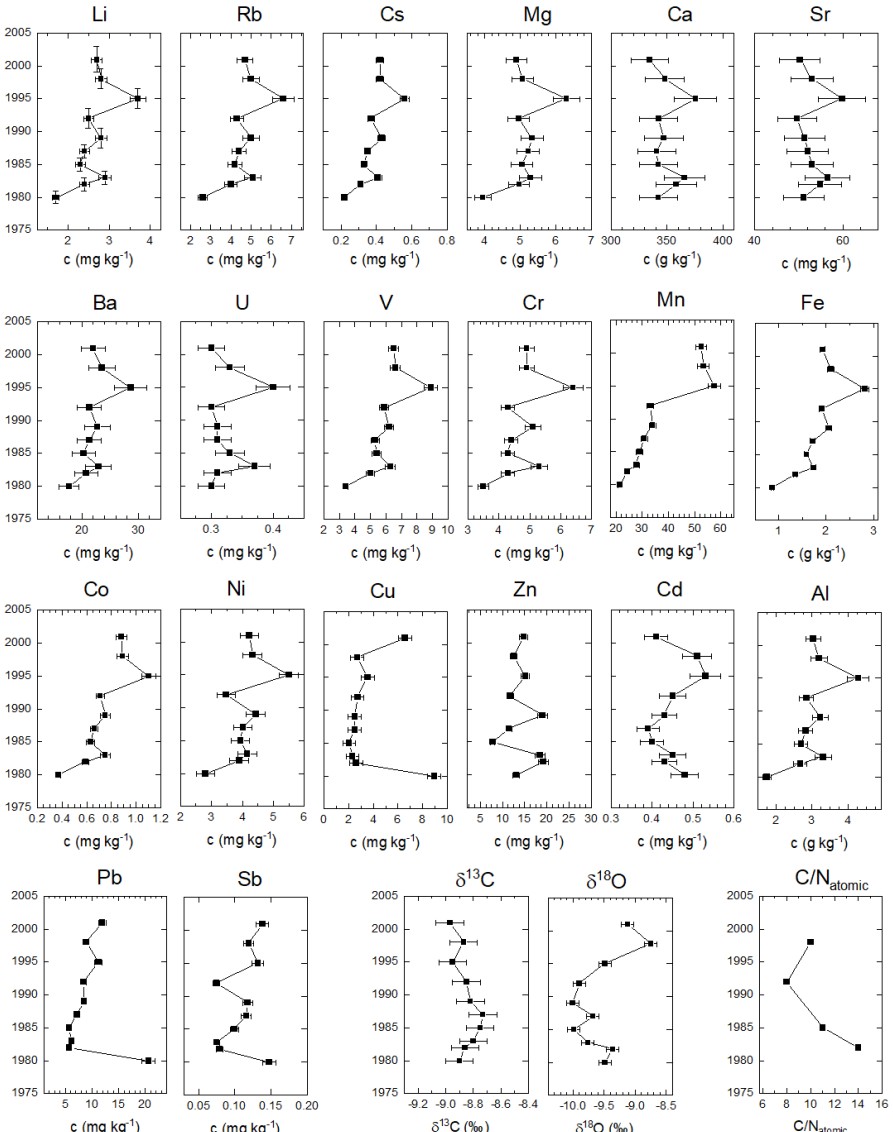

**Figure 3.** Mass concentrations of analyzed metals in tufa, $\delta^{13}C$ and $\delta^{18}O$ of tufa carbonate, and C/N ratio of organic part in tufa from the Korana River, presented in chronological order of layer deposition.

For most of the metals, a peak increase/change in mass concentration in the tufa was observed for 1995. The $\delta^{18}O$ values show change/peak in 1983 and in 1998, while the variations in $\delta^{13}C$ are within the measurement uncertainty. The C/N ratio was measured with a lower resolution (only four layers were analyzed). It ranges from 8 to 14, with the lowest values in 1993 and the highest in 1980. It has a general trend of decrease towards the outer tufa layers.

## 4. Discussion

### 4.1. Relationship between the Elements in Tufa

To explore geochemical characteristics and the relationships between the elements in tufa of the Korana River, a correlation analysis based on the symmetrical balances was carried out [58]. Following the hierarchical clustering, elements were assembled into two large clusters, which in turn were subdivided into two subclusters (Figure 4). The first included elements associated with the predominant authigenic carbonate fraction (Cd, U, Ca, Sr, Mg and Ba) and those of possible anthropogenic origin (Sb, Cu and Pb). Previous studies have indicated similar patterns for Sb and Pb in the Plitvice Lakes and thus possibly the same sources [45,46]. Among the main carbonate constituents, Ca shows stronger

clustering with Sr than with Mg. Furthermore, weak correlations of Mg with important elements of the aluminosilicate fraction, Cr and Al, may also indicate that part of Mg can be of terrigenous origin [26]. This is corroborated by mineralogical analyses (Appendix A, Figure A1) which show the presence of dolomite, probably as an allochtonous material. The closer clustering of Cd and U with Ca and Sr suggests that they are preferentially incorporated into the calcite structure [14,61].

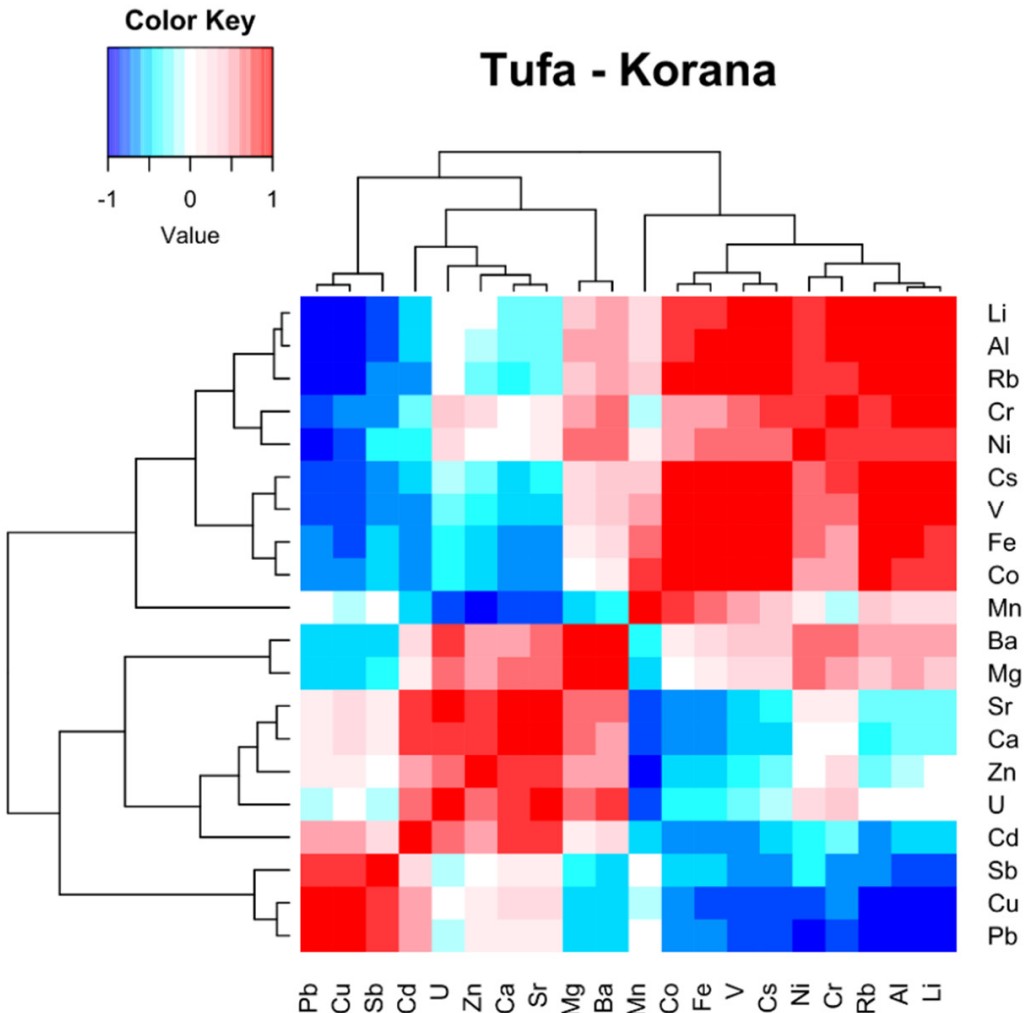

**Figure 4.** Heatmap of chemical elements in tufa of the Korana River based on symmetric coordinates [58]. The rows and columns were rearranged (with hierarchical clustering based on Ward's method) to cluster elements with similar geochemical behavior. The small graph in the upper left corner indicates the strength of the correlation. The red squares stand for a positive correlation and the blue squares for a negative correlation.

Heatmap results also show a strong gathering of elements (Li, Al, Rb, Cr, Ni, Cs, V, Fe and Co) predominantly hosted in clay minerals, oxyhydroxides and organic matter [62,63]. Here, only Mn is separated from this lithogenic cluster, but is close with Co and Fe, which may suggest the presence of Fe-Mn oxyhydroxides and their similar geochemical behavior, i.e., redox-sensitive nature [17,18].

### 4.2. Variations of Element Concentrations and of Stable Isotope Ratios

Concentrations of most of the analyzed elements in the Korana tufa are very low and reflect the dominance of calcite in the mineral composition which is also supported with X-ray powder pattern of sample 8, taken as a representative sample (Appendix A, Table A1).

Based on the average concentration, the elements analyzed are found in the following order: Ca > Mg > Al > Fe > Sr > Zn > Mn > Ba > Pb > V > Cr > Rb > Ni > Cu > Li > Co > Cd > Cs > Sb. Calcium concentration in all samples (334–375 g kg$^{-1}$) exceeds the average concentration in carbonate rocks [64], but is lower than the Ca concentration (~400 g kg$^{-1}$) in pure calcite. Concentrations of Mg are much lower (3.9–6.3 g kg$^{-1}$) and suggest the hypothesis about minor dolomite influence on tufa formation. Low Mg content in tufa is also corroborated by XRPD analyses which show a very small magnesium content in authigenic calcite (up to 1.5 mol%) and only traces of detrital dolomite (0.4–1.0 wt.%). Obtained results are in accordance with the fact that most freshwater carbonates and tufa have a low Mg concentration (and with references [10,11]), although there are examples of high-Mg calcite in freshwater sediments (e.g., in Lake Balaton [65]). Similar results on magnesium content in calcite were also obtained by Stoffers [66] for all but one of the samples analyzed from Plitvice Lakes. In the <2 µm fraction of this sample, which came from a very shallow small pond in Novaković a Brod, the latest and lowest of the Lower Lakes, he reported an asymmetrical calcite peak indicating 5 mol% $MgCO_3$ in solid solution. He explained this occurrence by the fact that the temperature and Mg/Ca ratio of the water increased from the Upper to the Lower Lakes, resulting in ratio values in some shallow parts of the Lower Lakes that favor the precipitation of Mg calcite. In contrast, the sampling site of this work is located below the confluence of the Plitvica Stream, which contains very cold water with a lower Mg/Ca ratio, causing a decrease in water temperature and Mg/Ca ratio in the outflowing of the Korana River. As a result, calcite with low magnesium content precipitates.

Concentrations of Sr and Ba do not vary significantly within samples; Sr ranges from 49.6 to 59.8 mg kg$^{-1}$ and Ba from 17.7 to 28.7 mg kg$^{-1}$. According to Teboul et al. [67], this range of concentrations is indicative of pure limestone sources of elements involved in the formation of tufa and suggests limestone dissolution in superficial conditions.

Concentrations of Cr, Cd and Ni, potentially toxic elements, do not show an increasing trend towards the more recent period. The average concentrations of Cd and Cr are 0.45 and 4.7 mg kg$^{-1}$, respectively, which is lower compared to upstream sites in the Kozjak Lake [45]. Compared to Cr, Ni also displays a similar trend and an average value of 4.1 mg kg$^{-1}$. According to Turekian and Wedepohl [64], the Cd concentration in average carbonate sediments is somewhat lower (0.35 mg kg$^{-1}$), while the concentration of Cr and Ni is two and three times higher (11 and 15 mg kg$^{-1}$) than in the Korana tufa.

In contrast, the concentrations of Zn, Pb, Sb and Cu are similar to those in the sediments of Lake Kozjak, with Pb, Sb and Cu increasing in the last layer. Zinc concentrations range from 7.8 to 19.2 mg kg$^{-1}$, while concentrations of Pb and Cu vary from 11.9 to 20.6 mg kg$^{-1}$ and from 2.1 to 8.9 mg kg$^{-1}$, respectively. Compared to the average concentrations in carbonates (Zn 20 mg kg$^{-1}$, Pb 10 mg kg$^{-1}$, Cu 4 mg kg$^{-1}$), Pb and Cu have higher concentrations in the Korana tufa (the concentrations are similar in Cukrov et al. [68]). The average Sb concentration is 0.11 mg kg$^{-1}$, significantly lower than the value of 0.47 mg kg$^{-1}$ in the upstream Kozjak Lake [46].

Cukrov et al. [68] analyzed the metal concentration and $\delta^{13}$C and $\delta^{18}$O in a tufa sampled from a pillar of the same bridge in the Korana River in the same time period. The chronology of their tufa was determined by counting laminae, and each of the 24 laminae were sampled and analyzed separately. They identified two concentration peaks for Cr, Mn, Fe, Co, Zn, Cd, Al and Pb, the first in 1983 and the second in 1993, while our work records the highest concentrations in 1995 and 1998.

The tufa mean value $\delta^{13}$C = −8.9 ± 0.1‰ (−8.8 ± 0.2‰, [68]) does not differ from the stable isotope composition of other tufa or lake sediment samples from the same area [36,37] (Figure 5). $\delta^{13}$C of dissolved inorganic carbon (DIC) measured near the tufa location (Lake Novaković a Brod) was −10.4 ± 0.5‰ (from −11.1 to −9.5‰, measured from 2003–2007, [69]). The fractionation shift is 1.5‰, which implies near equilibrium conditions for precipitation. This indicates that the tufa has been formed by bioinduced carbonate precipitation where most of the carbon from carbonate is precipitated from

DIC. Additionally, the range of the C/N ratio from 8 to 15 corresponds to the C/N of periphyton [40] that is usually a constituent of bioinduced tufa. The higher C/N values correspond to more cellulose-like plants (i.e., terrestrial and higher plants), while the lower ones, such as the observed value, correspond to aquatic algae.

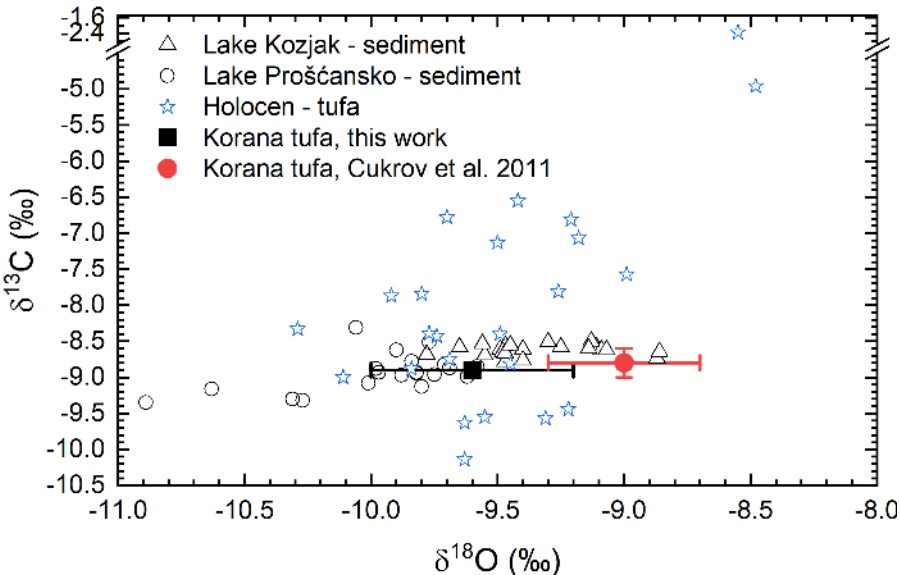

**Figure 5.** $\delta^{13}$C vs. $\delta^{18}$O for the Plitvice Lakes sediments from Lake Kozjak and Lake Prošćansko, for the Holocene tufa [36] and for tufa from the Korana River (data from this work and from Cukrov et al. [68]).

Tufa carbonate $\delta^{13}$C is largely influenced by the increase in $\delta^{13}$C of DIC, from which the carbonate precipitates. Because carbon from DIC exchanges with carbon from atmospheric $CO_2$ [70,71], there is an increase in $\delta^{13}C_{DIC}$ in the downstream direction, i.e., from Lake Prošćansko sediment to Lake Kozjak. The $\delta^{13}$C of tufa carbonate is similar to that of Lake Kozjak sediment.

The $\delta^{18}$O data of the tufa carbonate fraction (mean values $-9.6 \pm 0.4‰$, and in Cukrov et al. [68], $-9.0 \pm 0.3‰$) fit with $\delta^{18}$O for carbonates from Lake Kozjak sediment. The increase in $\delta^{18}$O from Lake Prošćansko to Lake Kozjak sediments reflects the increase in water evaporation in the downflow direction, i.e., from Lake Prošćansko to Lake Kozjak [72].

Some differences in the mean values of the $\delta^{13}$C and $\delta^{18}$O between the values presented here and those presented in Cukrov et al. [68] (Figure 5) can be attributed to the layer sampling strategy and the frequency of layer sampling, i.e., to the sampling resolution.

4.2.1. Manganese Concentration as an Indicator for Decrease in Concentration of Dissolved Oxygen

Manganese concentration in the analyzed samples ranged from 21.5 to 57.5 mg kg$^{-1}$, which is much lower compared to average limestone (700 mg kg$^{-1}$) or shale (850 mg kg$^{-1}$) concentrations [73]. It is well known that Mn as a redox-sensitive element has an important role in the transfer of trace elements from water to sediments [18,74]. In an oxygenated environment just below the oxic surface layer, manganese is presented in oxidation states Mn(III) and Mn(IV), forming insoluble Mn-oxyhydroxides that can be dissolved in reductive conditions to a soluble Mn(II) state [75], and afterward recycled between these two environments. Manganese and some redox-sensitive elements (Fe, Co, and V), which behave similarly, were plotted relative to Ca as the main phase of the matrix (Figure 6). The graph shows an increase in the ratios, especially of Mn/Ca to the external part of tufa. Previous works [43,76] on the Plitvice Lakes have emphasized periodic events of anoxia at the sediment/water interface. This could also be an explanation in the case of the Korana River tufa. Under such conditions, Mn could be remobilized from the sediments by the

dissolution of oxyhydroxides, but thereafter Mn is available for further precipitation as an oxide mineral and can adsorb other redox-sensitive elements, leading to their similar trends in the Korana tufa [17,77]. A similar explanation for the enrichment of Mn in highly carbonate sediments was also given by Herndon et al. [78]. The authors attribute this to the dissolution of oxyhydroxides, which occurs at lower levels of oxygen, and the subsequent removal of Mn from the water column primarily by precipitation on settling calcite grains.

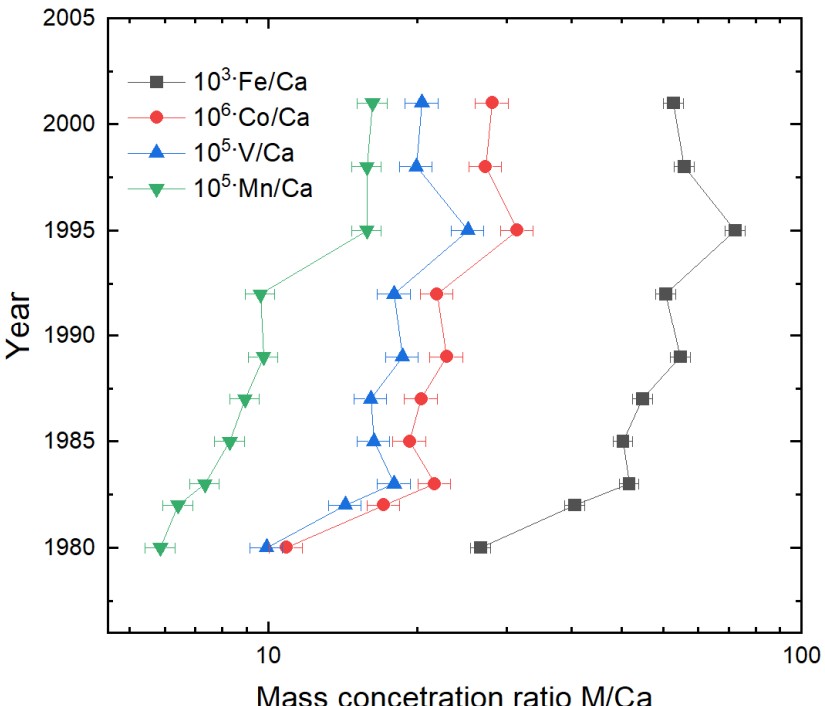

**Figure 6.** Mass concentration ratios of redox-sensitive elements (M): V, Mn, Fe and Co in relation to Ca (as the main constituent of the Korana tufa) in time.

### 4.2.2. Proxies for Water Temperature, Evaporation and Bioproductivity

When $\delta^{18}O$ is used as a temperature proxy, a lower $\delta^{18}O$ corresponds to higher temperature [24]. The $\delta^{18}O$ peak value appears in 1998 $\pm$ 1.5 (in 1996 [68]), and the lowest values are generally found in the period 1983–1995 (Figure 3). Th data set can be divided into three periods with distinctive $\delta^{18}O$ variations: (1) 1981–1983, where $\delta^{18}O$ is relatively constant (−9.4‰), (2) 1983–1995, where the lowest values of $\delta^{18}O$ are from −9.8‰, and (3) 1995–2003, where $\delta^{18}O$ increases to the highest value of −8.6‰. Then, using $\delta^{18}O$ in the temperature relation (in °C) [24]

$$T = 16.9 - 4.2\,(\delta^{18}O_c - \delta^{18}O_w) + 0.13\,(\delta^{18}O_c - \delta^{18}O_w)^2 \tag{4}$$

with $\delta^{18}O_c$ for calcite and $\delta^{18}O_w$ for water, and approximating $\delta^{18}O$ of water to −10.5‰ [72], the calculated temperature for period (1) is 12.4 °C, 14.0 °C for period (2), and for (3), it is 9.4 °C. However, an increase is observed in mean water temperature: from 12.3 °C (1982) to 12.7 °C (2012), and mean air temperature from 8.4 °C in 1980 to 9.7 °C in 2004 [8]. Lojen et al. [9], for tufa from the Krka River (Croatia), also did not find a correlation of carbonate $\delta^{18}O$ with temperature. For the use of $\delta^{18}O$ as a temperature proxy, one must also consider the $\delta^{18}O$ of the precipitation that also changes over time [72].

The changes in $\delta^{18}O$ of precipitated carbonates can also be influenced by limestone/dolomite detritus intake or evaporation effect. Both detritus carbonate intake (having $\delta^{18}O$ of 0‰) and evaporation [79], with their higher influence, result in a higher carbonate $\delta^{18}O$ value. The influence of detrital carbonate can be easily discarded since $\delta^{13}C$ values should correlate to $\delta^{18}O$, which they do not.

The variations in the second period must be observed with other data—$\delta^{13}C$ and major and trace metal concentrations.

The $\delta^{13}C$ increases from 1979 and peaks in 1987 (from $-9.0$ to $-8.7$‰) and then starts a general decline (Figure 3). The peak coincides with the minimum value of $\delta^{18}O$. No correlation between $\delta^{18}O$ and $\delta^{13}C$, except a negative correlation in the second period, can be interpreted as signal of an open system [80]. A positive correlation in laboratory conditions (only physical influences considered) was found to be linked with evaporation [81]. However, no correlation at all does not mean that evaporation does not take place. Therefore, it could be assumed that the change in $\delta^{18}O$ from (1) to (3) is due to an increase in the evaporation rate. The data in [68] also reveal some major intensive $\delta^{18}O$-related events between 1990 and 1996, when $\delta^{18}O$ values increase rapidly, reaching their maximum, which could also be linked to an intensive evaporation period. This coincides with a peak Mn/Ca value in 1995, which is connected to anoxic conditions and to local maxima in air temperatures in 1994 [8].

The highest $\delta^{13}C$ value coincides with lowest value for $\delta^{18}O$ around 1987, which could imply a somewhat stronger local primary production and/or intensive physical calcite precipitation. The introduction of geological carbonate (limestone) can be ruled out since it would move both $\delta^{13}C$ and $\delta^{18}O$ to higher values. Additionally, close to 1987, in 1993, a minimum in C/N value appears (implying more periphyton concentration in organic matter), so 1987–1993 can be considered as a period with maximum primary production taking place. It is interesting to observe that in [68], two obvious $\delta^{13}C$ maxima in 1993 and 2001 can also be attributed to intensive carbonate precipitation, either due to local primary production or the physical precipitation of calcium carbonate, controlled by temperature. The 1993 $\delta^{13}C$ peak [68] coincides with a minimum in $\delta^{18}O$ (Figure 3), while a $\delta^{18}O$ maximum in 1997 coincides with a local minimum in $\delta^{13}C$, implying two different effects taking place. Since the location considered is not a lake, but a flow channel (the beginning of the Korana River), the intensity of the water flow also must be taken in consideration, with their aspects of both physically and biologically induced precipitation. The 1993 $\delta^{13}C$ peak precedes the 1995 peak in major element concentration (e.g., Mg and Ca, Figure 3), and coincides (with a year delay) with their minimum (see also Cukrov et al. [68]).

The peaks in major element concentration in 1995 and in 1983 (Figure 3) could be linked to local maxima in air temperatures in 1994 and 1982 [8], and to a sudden decrease in discharge in 1983 (at Korana-Luketići station [81], near the position of the tufa growth). No data on water discharge are available for the period 1990–2001, so the peak value of 1994 cannot be attributed to a change in discharge, as with that of 1983. This effect could favor autochthonous tufa growth directly from the precipitation of dissolved inorganic carbonate from water to growth from the deposition of detrital mineral material suspended in water. This corresponds to the lowest $\delta^{13}C$ values, implying the lowest primary production [25], and calcite formed in tufa at this time was probably prevailingly originated from physical precipitation. This would be in accordance with Ihlenfeld et al. [14], where, inversely, a drop in Sr and Ba concentration (to their distinctive origin related to the surrounding lithology) was linked to high rainfall and so to their dilution.

Mg/Ca has a general ascending trend from 1980 to 2001 (as it is in accordance with Mg/Ca as a temperature proxy [9]), but not Sr/Ca, which is sometimes also considered as a temperature proxy (Figure 7). Mg/Ca of water, measured in periods 1980–1984 and 2010–2014, also showed a strong correlation with temperature [8]. However, Mg/Ca in the tufa does not seem to reflect the Mg/Ca decrease in water [8] between the two mentioned periods, when $HCO_3^{-}$ and $Ca^{2+}$ concentrations had increased by 15% and 7%, respectively, while $Mg^{2+}$ concentration stayed the same. Since the end of the period 2010–2014 is about a decade after the latest tufa layer, the Mg/Ca water value obviously points to a sudden change in geochemistry rather than a steady reflection of changes in environmental conditions. Additionally, concentrations of magnesium are fairly constant, while calcium and carbonate ions are the ones showing a large increase, implying some significant change in hydrology between 2003 and 2010.

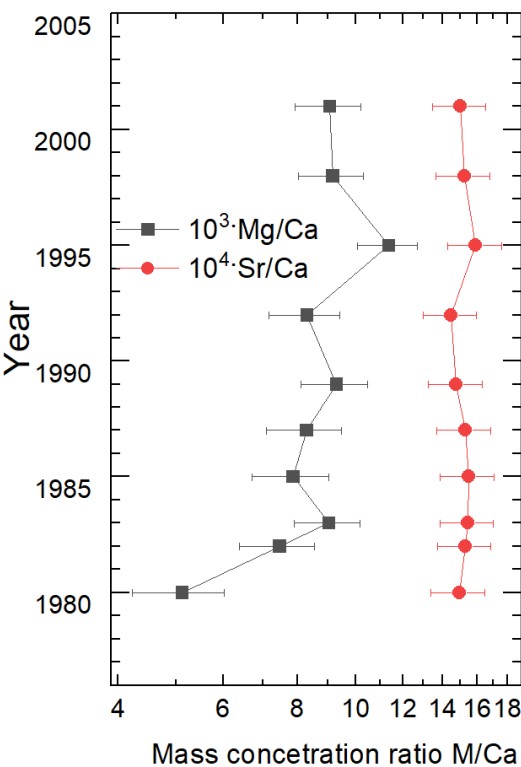

**Figure 7.** Change in mass concentration of Mg/Ca and Sr/Ca ratios in the Korana River tufa layers over time (M = Mg, Sr).

There are no data on water temperature for the observed period from 1980 to 2003, so the temperature change was approximated with air temperature from Sironić et al. [8], assuming the direct influence of air temperature to water temperature, implied also in [8]. The slope Mg/Ca vs. temperature is 0.0043 °C$^{-1}$, (R$^2$ = 0.54, p = 0.02), corresponding well to the slope of 0.0053 $\pm$ 0.0014 °C$^{-1}$ found for the Krka River (Croatia) water about 150 km away from the investigated area [9]. This implies that Mg/Ca in tufa can be used as temperature proxy. However, two factors argue against Mg/Ca as a proxy in this case. The first is the result of mineralogical analysis and element clustering (Figure 4) that implies that a part of Mg in tufa is alochtonous and so not temperature-dependent, and not relevant as a temperature proxy [26]. The other factor is that for number of major and trace elements to Ca ratio (X/Ca), correlation with temperature (Appendix A, Table A3) is significant ($p < 0.05$) and positive implying other causes and maybe only a coincidental relation to temperature. The correlation X/Ca vs. temperature was not found for Sr, U, Zn, Cd, Pb or Cu, while it was found for Ba, V, Cr, Mn, Fe, Co, Zn, Al, Ni and all alkali metals. The same is true also for the M/Al correlation vs. temperature, and consequently for M/Ca vs. Al/Ca ratios. The assumption that Mg/Ca is a temperature proxy is based on fact that the solubility of MgCO$_3$ decreases more strongly with temperature compared to that of CaCO$_3$; however, the solubility of, e.g., BaCO$_3$ actually increases with temperature [82], which is one of the examples where Ba/Ca shows a significant correlation with temperature. Therefore, other effects have to be considered, such as lower water discharges coincidental to temperature increase, which would concentrate elements later introduced to the system. This is the case of water-soluble alkali elements, other elements connected to the non-carbonate fraction, and detrital fractions (such as allochthonous dolomite containing Mg), probably brought later on by the water runoff. In general, it means that at certain conditions (at higher temperature and lower discharges), the influence of detrital/allochthonous elements are much higher, so using this principle, such conditions can be identified in tufa.

The fact that Sr, U and Cd do not exhibit a significant correlation with temperature can be explained by their properties to coprecipitate with calcite [83,84], while Cd, Cu, Pb

and Zn are scavenged by Ca [43]. The second group of elements is also discussed later in light of anthropogenic influence.

### 4.2.3. Synthesis of Chronological Changes through Analysis of Elemental and Stable Isotope Composition

The diagram in Figure 8 is an attempt to compare site measured parameters with elemental and isotope composition in the tufa and to explain synchronizing processes deducted from the tufa data. Throughout the period that the tufa precipitated, 1979 to 2003, there was an increase in the air temperature, decrease in water discharge until 1990 [81], and a significant decrease in Mg/Ca ratio in water between the periods 1980–1984 and 2010–2014 [8]. The increase in Mg/Ca in water, not observed in tufa, implies some event that took place in 2003–2010 (the period between the tufa being removed from water and the beginning of the second phase of water sampling) that lead to a change in hydrology. Peaks in metal concentrations and X/Ca were attributed to a drop in discharge and peak in temperature leading to their increased concentration. A decrease in oxygenation (proxy Mn/Ca) throughout the period, with a stronger decrease from 1995, was coincidental to increased evaporation (proxy $\delta^{18}O$). A simultaneous decrease in C/N with an increase in $\delta^{13}C$ was associated with stronger bioproductivity (1992–1995); however, an increase in $\delta^{18}O$ (associated with increase in evaporation) and a $\delta^{13}C$ peak in 2001 [68] imply more intensive physical precipitation on account of biologically induced precipitation.

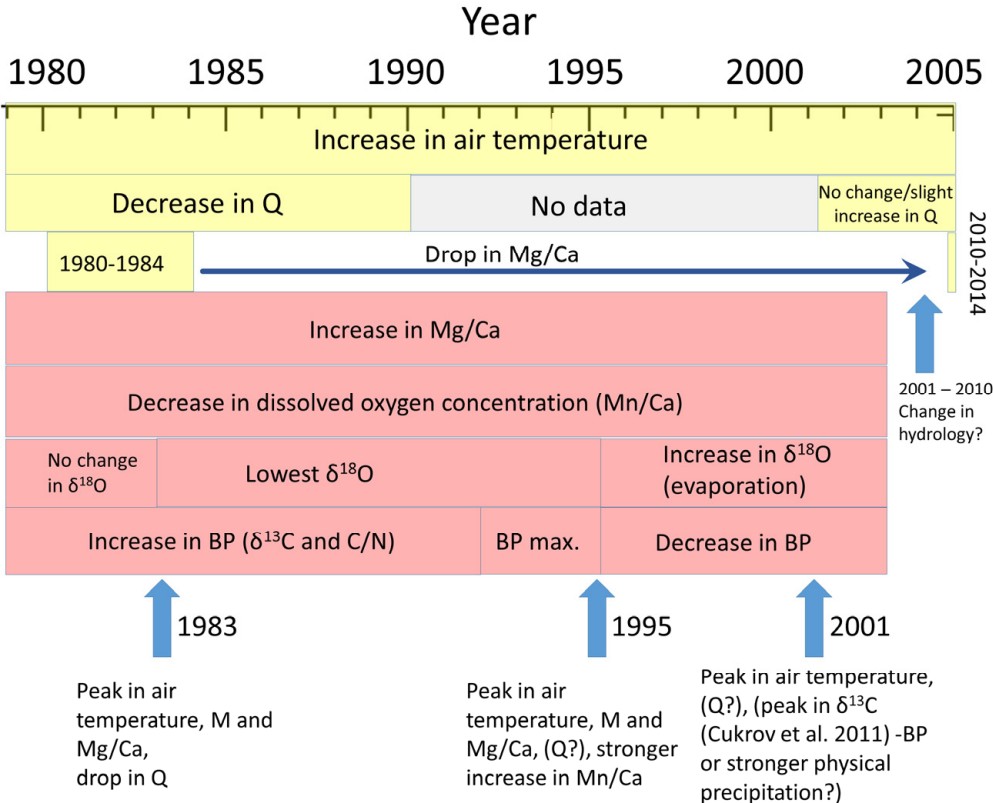

**Figure 8.** Chronological diagram comparing measured air temperature, water discharge (Q) [81], and Mg/Ca ratio in water [8] (in yellow) to elemental and isotope composition in the tufa (in pink), with presumable processes taking place (BP—bioproduction, M—element concentration, Cukrov et al. 2011 [68]). Blue arrows point to the peak events in the Korana River tufa.

This implies that in the research area (the water around tufa, and the tufa itself), environmental changes such as higher air temperature and a decrease in water recharge cause a decrease in dissolved oxygen concentration as well as an increase in bioproductivity, which can, however, with increased evaporation, lead to a decrease in bioproductivity on account of physical calcite precipitation.

### 4.3. Determination of the Anthropogenic Influence

To assess a possible anthropogenic influence on the chemical composition of the Korana tufa, the local enrichment factor (LEF) was used (Figure 9, Appendix A, Table A4). The LEF concept is useful to overcome the dilution effect caused by the predominant matrix phase, such as calcium carbonate, and it reflects the study-specific local background for the elements under investigation. Aluminum was used here as a reference element because (i) it has the best linear fit with suspected anthropogenic elements, (ii) it is abundant in clay minerals, and (iii) it is minimally affected by redox processes.

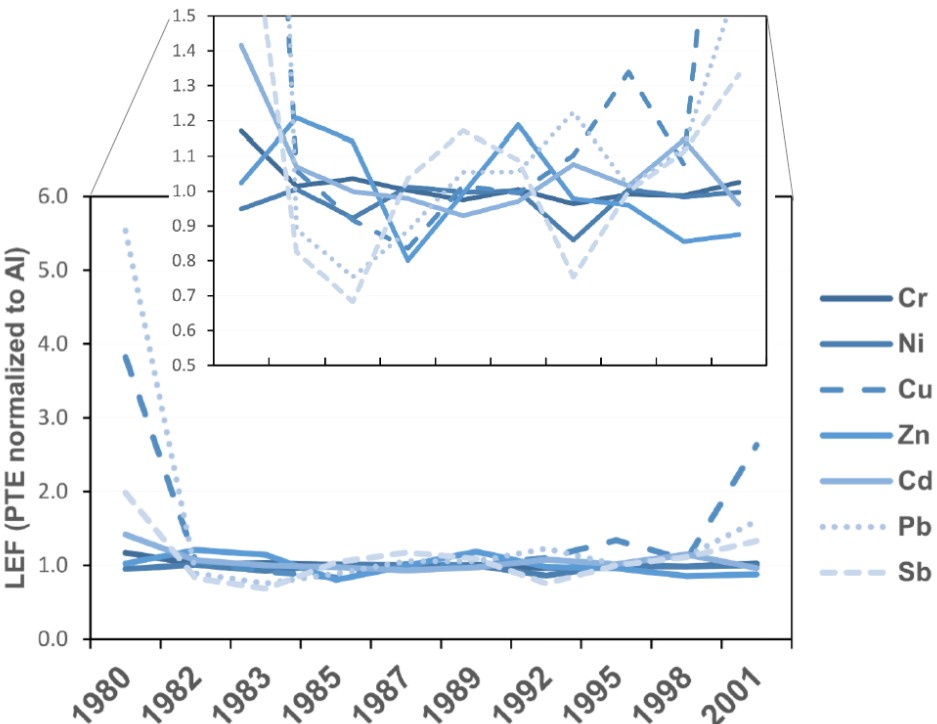

**Figure 9.** Local enrichment factors (LEFs) of potentially toxic elements relative to aluminum (Al) in samples of the Korana tufa. The upper diagram represents the enlarged lower diagram in the range of LEFs from 0.5 to 1.5 for better visibility. The elements are arranged following the periodic table groups.

We focus here only on seven elements that can be considered potentially toxic, namely Cr, Ni, Cu, Zn, Cd, Pb and Sb (Figure 9). The LEFs in the background samples of the Korana tufa are in the following ranges for the following elements: Cr, 0.95–1.17; Ni, 0.86–1.01; Cu, 0.94–1.34; Zn, 0.80–1.05; Cd, 0.93–1.15; Pb, 0.75–1.22; and Sb, 0.68–1.18. Chromium, Ni and Zn had no enrichment indicating that they are mostly geogenic in nature. Similar patterns of behavior for Cr and Ni are probably due to their presence in the clay minerals and oxyhydroxides. Zinc clustering with carbonate constituents (Figure 4) and the lower concentration than in limestones might suggest that it originates mainly from carbonate sources [73].

The enrichment patterns of Cd, Sb, Pb and Cu suggest a somewhat different nature and are probably influenced by human activities. However, the highest LEFs (Cu 3.82, Cd 1.42, Pb 5.53, Sb 1.98) observed in the first sample could also be due to contamination during subsampling and should be considered with caution. Recently, it has been shown [45] that the middle parts of the Plitvice Lakes sediment cores from 1950–2000 show an increasing trend in Pb concentrations, which has also been observed in tufa [68]. This increase was explained by expanding tourist and traffic activities that started after the World War II. After the 2000s, a slight decrease in Pb concentration can be observed in Lake Prošćansko, but for Lake Kozjak, this decrease was not found. Vukosav et al. [44] also suggest that the

elevated Pb concentrations are most likely a result of soil drainage from the Rječica tributary, which runs along the motorway and flows into Lake Kozjak. Similar to Pb, Bačić et al. [46] also found higher Sb enrichments in the upper part of the upstream sediment core in Lake Kozjak, dating from more recent times. Since Sb reflects similar behavior to Pb in tufa samples, it has probably the same anthropogenic sources related to traffic activities. The highest LEF was found for Cu in the most recent sample. According to the Plitvice Lakes National Park Management Plan [85], unresolved sewage systems and the permeability of the existing wastewater collection network represent a major pollution risk in the Park area. So far, many authors have highlighted a higher anthropogenic input of Pb and Cu from sewer pipe systems, domestic plumbing or the overuse of mineral fertilizers [52,86,87]. In addition, intensive building and expanding tourism makes another pollution threat that can cause higher yields of potentially toxic Pb and Cu downstream of the Plitvice Lakes.

## 5. Conclusions

We presented data showing major (Al, Ca, Fe and Mg) and trace (Ba, Cd, Co, Cr, Cs, Cu, Li, Mn, Ni, Pb, Rb, Sb, Sr, U, V and Zn) elements and stable isotope changes in tufa growing in the Plitvice Lakes from 1979 to 2003, in the time when a decrease in discharge rate and an increase in air temperature was observed.

The main conclusions are as follows:

(1) In all analyzed samples, almost stoichiometric calcite was the dominant phase. Such results on the magnesium content in calcite are in accordance with the fact that most freshwater carbonates and travertines have a low Mg content. This assumption was underpinned with the geochemical analysis, which also revealed very low Mg, Ba and Sr concentrations, indicative of pure limestone sources, and suggested limestone dissolution in superficial conditions. This was corroborated by $\delta^{13}$C data, with values typical for calcite precipitated from water ($-9.0$ to $-8.5$‰).

(2) Hierarchical clustering with distances based on the correlations indicated that the elements were grouped into two important clusters, which in turn were subdivided into two subclusters. The first cluster consisted of possible anthropogenic (Sb, Cu and Pb) and prevailing carbonate (Cd, U, Ca, Sr, Mg and Ba) subclusters. The second cluster consisted of a lithogenic fraction composed of Li, Al, Rb, Cr, Ni, Cs, V, Fe and Co, and a subcluster of Mn, which showed a redox-sensitive nature.

(3) Manganese increase towards the external part of tufa, reflected an increase in redox conditions (a decrease in oxygen levels from 1980 to 2001). Periodic events with increase in temperature, decrease in water discharge and peak in elements was observed in 1983 and with an increase in bioproduction in 1995. In 2001, a peak in precipitation resulting either from an increase in bioproduction or physical precipitation was observed.

(4) Mg/Ca in tufa did not reflect the measured decrease in water Mg/Ca. Mg/Ca as well as the most elements of the non-carbonate fraction correlated to the temperature increase over time; however, this was explained by lower water discharge and consequently increased concentrations of the elements. Using metal to Ca and Al ratios and observing peak events can be used to identify coincidental peaks in temperature and minima in water discharges. However, this still needs to be verified.

(5) $\delta^{18}$O indicated greater evaporation in the second half of the sediment (from $-9.8$ to $-8.6$‰). $\delta^{13}$C reflected bioproduction events, which decrease at lower discharges and greater evaporation.

(6) Enrichment patterns of Pb and Cu have highlighted their possible anthropogenic character in 1980 and 2001.

**Author Contributions:** Conceptualization, A.S., M.L. and I.F.; methodology, A.S., M.L. and I.F.; validation, D.T.; formal analysis, A.S. and M.L.; investigation, A.S., M.L. and D.T.; resources, A.S., M.L. I.F. and D.T.; data curation, A.S. and M.L.; writing—original draft preparation, A.S., M.L. and I.F.; writing—review and editing, D.T., A.S., M.L. and I.F.; visualization A.S., M.L. and I.F.; supervision, A.S.; project administration, M.L.; funding acquisition, M.L. and I.F. All authors have read and agreed to the published version of the manuscript.

**Funding:** This research was partially funded by Project 062-098 2709-0510 of the Ministry of Science, Education and Sports of Croatia.

**Institutional Review Board Statement:** Not applicable.

**Informed Consent Statement:** Not applicable.

**Data Availability Statement:** Not applicable.

**Acknowledgments:** The authors thank the Stable Isotope Laboratory SILab of the University of Rijeka and their four-year Project 062-098 2709-0510 for stable isotope measurements under the leadership of Zvjezdana Roller-Lutz, with analyses conducted by Magda Mandić. Authors thank scuba divers who collected the sample for this research; Branko Jalžić and Anđelko Novosel for providing data for Lake Kozjak; Ines Krajcar Bronić as the originator of the idea and valuable text corrections; and Jadranka Barešić for incisive advice.

**Conflicts of Interest:** The authors declare no conflict of interest. The funders had no role in the design of the study; in the collection, analyses, or interpretation of data; in the writing of the manuscript, or in the decision to publish the results.

## Appendix A

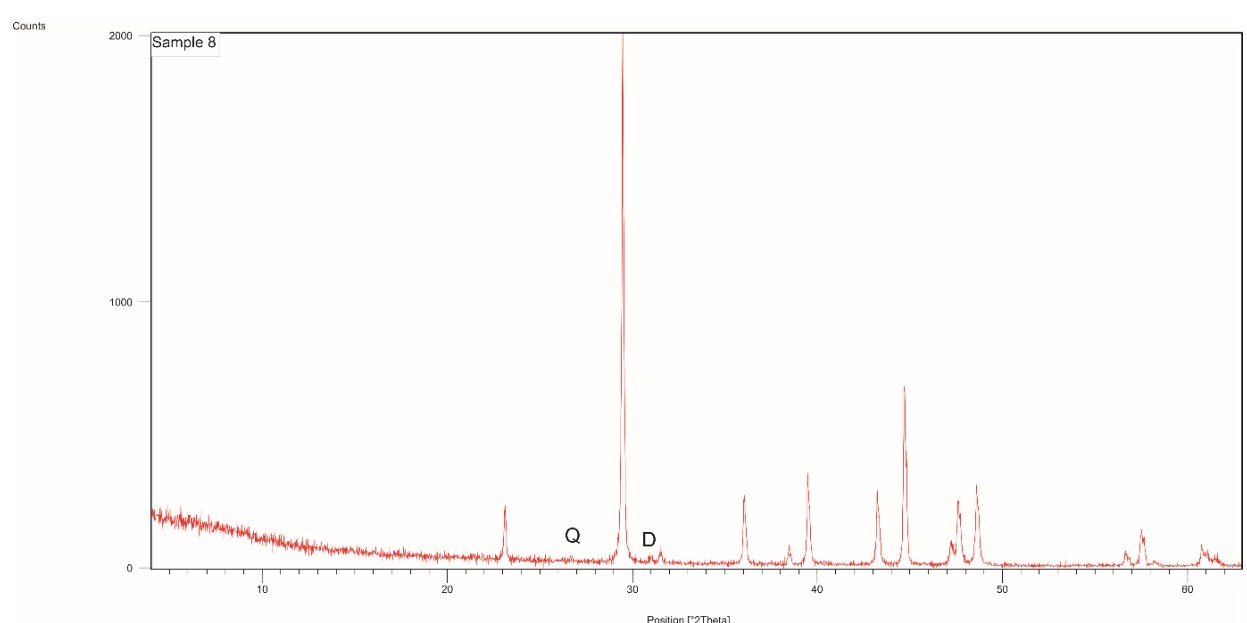

**Figure A1.** X-ray powder pattern of sample 8 (CuKα radiation). All peaks except those at 26.65 (quartz—Q) and 30.99 (dolomite—D) °2θ belong to calcite.

**Table A1.** Element concentrations (mg kg$^{-1}$), with relative uncertainties of measurement (%) in tufa samples. The elements are arranged following the periodic table groups.

| | | | | | | | | | | | | | | | | | | | |
|---|---|---|---|---|---|---|---|---|---|---|---|---|---|---|---|---|---|---|---|
| | | | | | | | **Element Concentration (mg kg$^{-1}$)** | | | | | | | | | | | | |
| **Sample Label** | **Li** 5% | **Rb** 8% | **Cs** 5% | **Mg/1000** 6% | **Ca/1000** 5% | **Sr** 9% | **Ba** 10% | **U** 7% | **V** 5% | **Cr** 5% | **Mn** 4% | **Fe/1000** 3% | **Co** 5% | **Ni** 7% | **Cu** 7% | **Zn** 6% | **Cd** 7% | **Al/1000** 7% | **Pb** 6% | **Sb** 7% |
| 1 | 1.7 | 2.6 | 0.22 | 3.94 | 342 | 51.1 | 17.7 | 0.30 | 3.4 | 3.5 | 21.5 | 0.86 | 0.37 | 2.80 | 8.93 | 13.1 | 0.48 | 1.76 | 20.6 | 0.15 |
| 2 | 2.4 | 4.0 | 0.31 | 4.97 | 358 | 54.8 | 20.7 | 0.31 | 5.0 | 4.3 | 24.3 | 1.35 | 0.59 | 3.90 | 2.60 | 19.2 | 0.43 | 2.68 | 5.7 | 0.08 |
| 3 | 2.9 | 5.1 | 0.41 | 5.29 | 366 | 56.5 | 22.9 | 0.37 | 6.3 | 5.3 | 27.8 | 1.74 | 0.75 | 4.16 | 2.32 | 18.4 | 0.45 | 3.30 | 6.2 | 0.07 |
| 4 | 2.3 | 4.2 | 0.33 | 5.05 | 342 | 53.0 | 20.3 | 0.33 | 5.4 | 4.3 | 29.0 | 1.58 | 0.63 | 3.94 | 2.05 | 7.8 | 0.40 | 2.69 | 5.7 | 0.10 |
| 5 | 2.4 | 4.4 | 0.35 | 5.21 | 341 | 52.0 | 21.3 | 0.31 | 5.3 | 4.4 | 30.8 | 1.71 | 0.66 | 4.02 | 2.51 | 11.5 | 0.39 | 2.82 | 7.2 | 0.12 |
| 6 | 2.8 | 5.0 | 0.43 | 5.34 | 347 | 51.3 | 22.7 | 0.31 | 6.2 | 5.1 | 34.0 | 2.06 | 0.75 | 4.43 | 2.51 | 19.0 | 0.43 | 3.23 | 8.5 | 0.12 |
| 7 | 2.5 | 4.3 | 0.37 | 4.96 | 343 | 49.6 | 21.3 | 0.30 | 5.9 | 4.3 | 33.1 | 1.91 | 0.71 | 3.48 | 2.73 | 11.7 | 0.45 | 2.84 | 8.4 | 0.07 |
| 8 | 3.7 | 6.6 | 0.56 | 6.30 | 376 | 59.8 | 28.7 | 0.40 | 8.9 | 6.4 | 57.5 | 2.82 | 1.10 | 5.50 | 3.56 | 15.2 | 0.53 | 4.27 | 11.2 | 0.13 |
| 9 | 2.8 | 5.0 | 0.42 | 5.06 | 348 | 53.0 | 23.5 | 0.33 | 6.6 | 4.9 | 53.3 | 2.10 | 0.89 | 4.32 | 2.71 | 12.5 | 0.51 | 3.19 | 8.9 | 0.12 |
| 10 | 2.7 | 4.7 | 0.39 | 4.89 | 334 | 50.2 | 22.0 | 0.30 | 6.5 | 4.9 | 52.5 | 1.93 | 0.88 | 4.22 | 6.57 | 14.8 | 0.41 | 3.03 | 11.9 | 0.14 |
| Mean | 2.6 | 4.6 | 0.38 | 5.10 | 349 | 53.1 | 22.1 | 0.33 | 6.0 | 4.7 | 36.4 | 1.81 | 0.73 | 4.08 | 3.6 | 14.3 | 0.45 | 2.98 | 9.43 | 0.11 |

**Table A2.** Stable isotope composition (‰) (uncertainty 0.05‰) and atomic C/N ratio for each analyzed layer.

| Sample Label | $\delta^{13}$C (‰) | $\delta^{18}$O (‰) | C/N$_{atomic}$ |
|---|---|---|---|
| 1 | −8.90 | −9.49 | n.a. |
| 2 | −8.86 | −9.36 | n.a. |
| 3 | −8.80 | −9.76 | 14 |
| 4 | −8.75 | −9.99 | n.a. |
| 5 | −8.73 | −9.68 | 11 |
| 6 | −8.82 | −10.01 | n.a. |
| 7 | −8.85 | −9.90 | n.a. |
| 8 | −8.95 | −9.49 | 8 |
| 9 | −8.87 | −8.75 | n.a. |
| 10 | −8.97 | −9.12 | 10 |
| Mean | −8.85 | −9.56 | 11 |

Note: n.a.—not analyzed.

**Table A3.** Significance of correlations of M/Ca and M/Al (M element) vs. temperature (T), M/Ca vs. Mg/Ca and M/Ca vs. Al/Ca for the Korana River tufa layers ("+" significant positive correlation, "-" no correlation, $p < 0.05$).

| Sample Label | Significance of Correlation for | | | |
|---|---|---|---|---|
| | M/Ca vs. T | M/Al vs. T | M/Ca vs. Mg/Ca | M/Ca vs. Al/Ca |
| Li | + | + | + | + |
| Rb | + | + | + | + |
| Cs | + | + | + | + |
| Mg | + | + | + | + |
| Sr | - | - | - | - |
| Ba | + | + | + | + |
| U | - | - | - | - |
| V | + | + | + | + |
| Cr | + | + | + | + |
| Mn | + | + | - | + |
| Fe | + | + | + | + |
| Co | + | + | + | + |
| Zn | - | - | - | - |
| Cd | - | - | - | - |
| Al | + | + | + | + |
| Pb | - | - | + | - |

**Table A4.** Background functions of potentially toxic elements (PTEs) in the studied tufa samples used to calculate local enrichment factors (LEFs). The best fit was obtained using a least trimmed squares (LTS) regression as a robust statistical method. The natural concentrations of the target elements (Cr, Ni, Cu, Zn, Cd, Pb and Sb) were predicted by empirical functions of Al as the independent variable. Background concentrations (EBN) were predicted by the formula: [intercept + (slope × Al)]. LEFs were then calculated by dividing the concentrations of the PTEs by their predicted values (LEF = E/EBN). The elements are arranged following the periodic table groups.

| Element | $R^2$ | Intercept | Slope |
|---|---|---|---|
| Cr | 0.96 | 0.56336 | 0.00139 |
| Ni | 0.92 | 1.19887 | 0.00100 |
| Cu | 0.70 | 2.11433 | 0.00013 |
| Zn | 0.49 | 2.12011 | 0.00322 |
| Cd | 0.65 | 0.21260 | 0.00007 |
| Pb | 0.63 | −1.41657 | 0.00292 |
| Sb | 0.50 | 0.03438 | 0.00002 |

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
