# Peer review of "Environmental Changes Recorded in Tufa from the Korana River, Croatia: Geochemical and Isotopic Approach"

_water, doi:10.3390/w15071269_

Round 1

Reviewer 1 Report (Previous Reviewer 3)

The manuscript a study of the chemical composition and isotopic composition of O, C in a recent laminated tufa developed around a wooden pillar on the banks of Plitvice Lakes from 1979 to 2003. This can be important to find a possible anthropogenic contribution of potentially toxic elements.

This manuscrkipt has been previously submitted to Water and now it has been highly improved. I provide only very small suggestions to change.

Line 152. “Tufa, analyzed in this paper, was located in the area built “. Replace “was located in..” by  “is located in…”

Figure 1.  The direction of the cross-section should be a straight line from A to E.

At the end of the figure caption there is an extra parenthesis.

Line 365mg kg-1, which

Line 309-326. Use present tense instead of past tense.

Author Response

Best regards, Andreja 

Reviewer 2 Report (New Reviewer)

Dear authors,

Please see attachment file.

Best wishes.

Author Response

Please see the attached file. Thank you very much for your valuable comments.

Best regards,

Andreja

This manuscript is a resubmission of an earlier submission. The following is a list of the peer review reports and author responses from that submission.

Round 1

Reviewer 1 Report

Page 10 Line 329:

Manganese concentration in analyzed samples ranged from 21.5 to 57.5 mg/kg which is much lower compared to the average limestone (700mg/kg)

Author Response

Please find the response in the attached file.

Reviewer 2 Report

See attached.

Author Response

(The authors gave the same response as above.)

Reviewer 3 Report

The manuscript a study of the chemical composition and isotopic composition of O, C in a recent laminated tufa developed around a wooden pillar on the banks of Plitvice Lakes from 1979 to 2003. This can be important to find a possible anthropogenic contribution of potentially toxic elements.

The article looks very interesting. It is a pity that the study was only carried out on a sample.

The manuscript. Some grammatical corrections are necessary and other moderate changes needed.

I suggest some other minor changes:

Line 34: formed by physical and bioinduced precipitation “  by??

Line 40: “  Tufa de-posited in layers can be used as archive of fast changes”. Complete this sentence “fast changes of….”

Line 89: this is not the correct place for references [48, 49]. In addition reference [48] is highly incomplete ???

Line 89: (i) to observe.. it is not an observatikon perphas it is better to use the term: “to determine”.

Figure 1. Could you indicate the direction of the geoilogical profile on the geological map?

Figure 9. The Y-axis could be between 0 and 3. This would allow a better appreciation of the variations in metal content.

The quality of this figure must be improved.

Line 321-322: “The in crease in δ18O from Lake Prošćansko to Lake Kozjak sediments reflects increase in water  evaporation on downflow direction, i.e. from Lake Prošćansko to Lake Kozjak”. Other explanations could be possible, please read more about this issue.

Figure A1. Indicate on the diffractogram to which phase each of the most important peaks corresponds.

In conclusion, I believe that the conclusions, while appearing interesting, would be more robust if they were supported by a larger sample. Many of them are based on previous results.

Author Response

(The authors gave the same response as above.)

Round 2

Reviewer 2 Report

See attachment.
